Manuscript prepared for Earth Surf. Dynam.
with version 2015/04/24 7.83 Copernicus papers of the LaTeX class copernicus.cls.
Date: 8 November 2016

# Graffiti for science - Erosion painting reveals spatially variable erosivity of sediment-laden flows

Alexander R. Beer[1,2], James W. Kirchner[2,1], and Jens M. Turowski[3]

[1]WSL Swiss Federal Institute for Forest, Snow and Landscape Research, 8903 Birmensdorf, Switzerland
[2]Department of Environmental System Sciences, ETH Zurich, 8092 Zurich, Switzerland
[3]GFZ German Research Centre for Geosciences, Telegrafenberg, 14473 Potsdam, Germany

*Correspondence to:* A. R. Beer (alexander.beer@wsl.ch)

**Abstract.** Spatially distributed detection of bedrock erosion is a long-standing challenge. Here we show how the spatial distribution of surface erosion can be visualised and analysed by observing the erosion of paint from natural bedrock surfaces. If the paint is evenly applied, it creates a surface with relatively uniform erodibility, such that spatial variability in the erosion of the paint reflects variations in the erosivity of the flow and its entrained sediment. In a proof-of-concept study, this approach provided direct visual verification that sediment impacts were focused on upstream-facing surfaces in a natural bedrock gorge. Further, erosion painting demonstrated strong cross-stream variations in bedrock erosion, even in the relatively narrow (5 m wide) gorge that we studied. The left side of the gorge experienced high sediment throughput with abundant lateral erosion on the painted wall up to 80 cm above the bed, but the right side of the gorge only showed a narrow erosion band 15 - 40 cm above the bed, likely due to deposited sediment shielding the lower part of the wall. This erosion pattern therefore reveals spatial streambed aggradation that occurs during flood events in this channel. The erosion painting method provides a simple technique for mapping sediment impact intensities, and qualitatively observing spatially distributed erosion in bedrock stream reaches. It can potentially find wide application in both laboratory and field studies.

## 1 Introduction

Fluvial bedrock erosion is an important control on stream channel development (and thus on whole landscape evolution) in steep mountainous terrain and tectonically active regions. Bedrock erosion in stream channels is driven by several interacting processes, of which the most efficient are hydraulic shear detachment of weak bedrock, plucking of bedrock blocks, and abrasion of small bedrock grains due to sediment impacts. Dissolution and cavitation can also be important contributors to bedrock erosion under specific conditions (Whipple et al., 2000; Sklar and Dietrich, 2004; Lamb et al., 2008; Vachtman and Laronne, 2013). Bedrock topographic features, together with the interplay of the sediment tools and cover effects (impacting sediment act as erosive tools while stationary sediment

can protect surfaces against impacts), regulate the rate and spatial pattern of local surface erosion (Gilbert, 1877; Sklar and Dietrich, 2004; Turowski et al., 2008; Johnson et al., 2009; Yanites et al., 2011; Cook et al., 2014; Beer et al., in review).

  Spatially distributed measurements of natural bedrock erosion rates are valuable for understanding the underlying process physics, as well as for modelling landscape evolution and designing engi-
neered structures. Repeated measurements of local or reach-scale rates of vertical erosion (i.e. channel incision), lateral erosion (channel widening) and downstream-directed erosion of protruding bedrock surfaces are needed to better understand bedrock channel evolution. However, quantifying spatially distributed bedrock erosion rates in natural settings is challenging and few such measurements exist (e.g., Hartshorn et al., 2002; Stock et al., 2005; Turowski et al., 2008; Johnson et al.,
2010; Stephenson, 2013; Wilson et al., 2013; Cook et al., 2014; Beer et al., in review).

  Documenting subtle topographic changes in bedrock surfaces has typically required sophisticated instruments and techniques, including photogrammetry, total stations, laser scanners, and erosion meters (Turowski and Cook, 2016). A much simpler, albeit more indirect method, has hardly been considered yet: painting. Paint is commonly used in fluvial geomorphology to visualize and track
tracer particles (e.g., single bedload pebbles to analyse sediment transport; see overview by Hassan and Ergenzinger, 2003). However, it has rarely been used to study spatially distributed surface changes. Dietrich et al. (2005) and Surian et al. (2009) painted small patches of streambed sediments to study how sediment transport dynamics vary with channel characteristics. Gill and Lang (1983) applied several paint dots along shoreline bedrock platforms to get a general overview of erosion at
large spatial scales. To our knowledge, however, paint has not been used to analyse the local spatial distribution of erosion on natural bedrock surfaces.

  Here, we explore an easy, inexpensive method for monitoring spatial patterns of bedrock erosion, which we term erosion painting. We evaluate its applicability using a 3-year series of photographs of painted bedrock surfaces in a natural bedrock gorge in the Swiss Alps, and illustrate how this simple
method gives insight into sediment transport and erosion processes during high-flow events.

## 2 Methods

We present a proof-of-concept field study demonstrating the scientific potential of the following general approach. We used environmentally safe and water-insoluble latex-based dispersion paint to cover natural bedrock surfaces that were expected to show varying patterns of erosion (see below for
a description of the field site), and photographed these surfaces from defined vantage points regularly during visits to the sites. Comparisons of sequential photographs from the same vantage points were then used to document the removal of paint by erosive events. To compare specific details of interest over time, it was helpful to include retrievable features (benchmarks) in the pictures. The observed pattern of eroded and remaining paint indicates the spatial distribution of erosion. More precisely, to

the extent that the paint provides a uniformly erodible surface, we suggest that the spatial pattern of paint erosion reflects the spatial pattern in the erosivity of the flow and the sediment that it carries (i.e. their erosive strength or potential to erode the bedrock). For useful results to be obtained, this erosivity must be high enough to remove some of the paint, but also low enough that some paint remains.

The field site for this study was a 30 m-long and 5 m-wide semi-alluvial bedrock gorge of the Gornera glacial meltwater stream above Zermatt, Switzerland (Figure 1). The local bedrock is serpentinite, and the bed sediment consists of both serpentinite and gneiss. The gorge is regularly flushed with up to 3 m deep sediment-laden flows due to hydropower operations upstream (Figure 1 B). In between these flushing events of 15 - 30 min length each, there is negligible discharge in the gorge (Figure 1 C and D). Due to the characteristics of the flushing operations (i.e. short, steep hydrographs and evacuation of previously accumulated sediment), the mean transported bedload grain size ($D_{50}$) likely varies considerably during each flushing event and between flushing events. The $D_{50}$ of the natural streambed upstream of the hydropower water intake is $\sim$4 cm at low flows, but it is unknown whether the average sediment load (including high flows) is finer or coarser than this. The sediment bed surface in the gorge returns to roughly the same height following each flushing event, but it likely varies strongly during the flushings themselves (Beer et al., in review).

We repeatedly painted several patches of the gorge's bedrock surface over a period of 3 years and photo-documented the resulting spatial patterns of eroded paint, renewing the paint as needed. To visualize variations of erosion with height above the streambed, we painted several vertical stripes of 0.15 m width and 2.0 m height on two opposing straight and smooth bedrock walls, starting at the sediment bed surface (Figure 1 C and D; we unfortunately could not paint below the sediment surface due to standing water in the sediment body). On the left gorge wall we connected two of these vertical stripes by horizontal lines to create a simple staff gauge, acting as a reference for a water surface altimeter positioned above the gorge. For analysis of the spatial bedrock erosion distribution across the streambed, we further painted a 2.5 m$^2$ wall section that laterally protruded into the streamflow, as well as the 20 m$^2$ top surface of a smooth bedrock boulder and the 3.2 m$^2$ smooth upstream face of a vertical bedrock slab (Figure 1 C and D), both of which protruded from the streambed. We validated the inferred patterns of bedrock erosion by comparing photos of worn paint to contemporaneous quantitative erosion analyses based on repeated high-resolution terrestrial laser scanning surveys of the same surfaces (TLS; Beer et al., in review). We also compared paint erosion patterns on the opposing bedrock walls to draw inferences on spatial patterns of sediment transport during the flushings.

## 3 Results

Even over short periods (i.e. a few flushing events), paint erosion was visible over most of the studied bedrock gorge section. The painted stripes on the opposing smooth bedrock walls revealed different erosion patterns: On the left gorge wall, the painted staff gauge (cf. Figure 1 D) was completely eroded up to 0.8 m above the streambed during a first study period of nearly one month with 44 flushing events (Figure 2 A and B). The staff gauge's paint was not renewed, and in the following three-week study period, a comparable flushing series ran through the gorge (Figure 2 D). The pattern of the eroded paint in the second period changed only slightly compared to the one observed in the first period, revealing slow paint erosion above 0.8 m on the painted staff gauge (compare the right vertical paint stripe in Figure 2 B and C). The qualitative erosion pattern of the staff gauge's paint is consistent with the quantitative bedrock surface change detection TLS-data of Beer et al. (in review), which show decreasing erosion rates with height over the streambed at this location (Figure 2 E and F present average erosion rates over the longer time frame of 04 June 2012 to 08 August 2013, with around 200 flushing events of varying lengths, varying flushed volumes and probably varying grain size distributions, comprising the bulk of the erosive events in both these years). Relationships between these bedrock surface changes and paint erosion are further detailed in the discussion section.

On the right gorge wall, both painted vertical stripes R1 and R2 (cf. Figure 1 D) consistently and repeatedly indicated stable, spatially localized zones of paint erosion, as shown in Figure 3 for stripe R2. These zones of completely eroded paint were found roughly 15 - 40 cm above the streambed during dry conditions. Above and below this erosion band, the paint generally remained intact, but showed zones that were slightly eroded during periods with higher flushing frequencies or flushing intensities (compare the first and the second rows to the third row shown in Figure 3).

Characteristic spatial patterns of eroded paint were observed at the laterally protruding wall section, and at the boulder and slab protruding from the streambed (Figure 4; cf. Figure 1 C and D). The protruding wall section was predominantly eroded on its upstream-facing and upward-facing sides (Figure 4 A to B), i.e. on those faces most prone to sediment impacts. The eroded paint on the boulder showed spatial patterns that are typical for the formation of upstream facing convex surfaces (UFCS; cf. Richardson and Carling, 2005; Wilson et al., 2013): (i) vertical erosion on planar surfaces in line with the streambed (i.e. incision; Figure 4 C to D), (ii) downstream-directed erosion on upstream-facing regions with abundant impact marks (Figure 4 E and F), (iii) no erosion on downstream-facing regions with nearly no impact marks (Figure 4 E and F), and (iv) a distinct crestline separating both regions (Figure 4 E and F). The vertical-standing bedrock slab, which was overflowed by at least some of the flushings, revealed a spatially homogeneous pattern of downstream-directed erosion on its upstream face (Figure 4 G to H). Only a few small parts of the upstream face of the slab were not eroded, because they were oriented away from the general direction of streamflow and sediment transport. Slab surfaces facing laterally, upward and downstream did not show any paint erosion over all three years studied (cf. the inset in Figure 4 H).

## 4 Discussion and Conclusions

In the following, we first assess the erosion painting method based on our proof-of-concept study. We then use this technique to draw inferences about spatial erosion processes at our study site, and discuss potential future applications in the geosciences.

### 4.1 General assessment of the erosion painting technique

This study illustrates erosion painting as a straightforward technique for visualizing the spatial distribution of the erosivity of sediment-laden flows. The paint remained on bedrock surfaces that were frequently submerged, showing that it could resist fluvial shear detachment and water dissolution (see the inset in Figure 4 H). In contrast, the paint was removed from surfaces where frequent sediment impacts were likely (e.g., Figures 2 C, 3 B and 4 B). This sediment-driven paint abrasion was clearly evident on surfaces where patchy paint still remained (cf. the slight erosion zones in Figure 3 B, and the upstream-facing part of the crestline in Figure 4 E). The transient paint erosion on the higher parts of the staff gauge between Figure 2 A and C, and also the slight erosion zones above and below the zone of complete erosion in Figure 3 B, indicated regions with lower sediment impact frequencies. Hence, erosion painting provides a semi-quantitative measure of the spatial distribution of sediment impact intensity, i.e. the erosivity of the streamflow. Assuming that impacting grains that remove the paint also abrade the underlying bedrock (which is reasonable from Figure 2 E, and from the impact marks in Figure 4 F), the erosion painting procedure can be further considered as an indirect measure of bedrock erosion. However, it is only a qualitative indicator of bedrock erosion and does not allow quantitative inferences of bedrock erosion rates.

Erosion painting is inexpensive, requires no fixed installations (apart from the paint itself), is straightforward to implement even in challenging locations, permits quick high-resolution field surveys (requiring only visual inspection of the surfaces and reference photographs), and can detect even low levels of streamflow erosivity. However, drawing quantitative inferences on erosion rates would require calibration against independent measurements, because the erodibility of the paint and the underlying bedrock will typically differ by large factors (see further discussion below). Environmentally friendly paint should be used, and only small surface patches should be painted to limit paint consumption and the visual impact of the technique. Any necessary permission should be requested, particularly for sensitive field areas. The paint should be applied carefully (e.g., avoiding wet and dusty rock, and leaving sufficient time for drying), since incorporated air bubbles or insufficient drying could lead to shear detachment of the paint by flowing water alone, without abrasion of the surface.

## 4.2 Process inferences from erosion painting at the Gornera

The paint erosion pattern at the staff gauge (Figure 2 B and C) clearly indicated erosion by sediment impacts (i.e. the sediment tools effect), and its decreasing strength with height above the bed due to a decreasing concentration of abrasive tools (as discussed by Fuller et al., 2016; Beer et al., in review). Below 0.8 m, erosion was strong enough to completely remove the paint during the first study period (Figure 2 A and B). Paint erosion at this level also reflects the slight inclination of the wall, resulting in surfaces that face slightly upward. Here, erosive sediment impacts from deflected grains falling through the water column have likely driven lateral erosion (cf. Fuller et al., 2016; Beer et al., in review). Above 0.8 m, fewer sediment impacts due to lower sediment concentrations could be inferred from the incomplete removal of paint over both study periods (Figure 2 A - C), indicating the low erosivity of the flow at these heights above the bed.

Quantitative TLS-based spatial bedrock erosion measurements (over two years with more than 200 flushing events of various discharges, length and volumes; see Figure 2 E) confirmed the decrease in sediment impacts with height above the bed, as qualitatively inferred from erosion painting. The uncertainty in the individual TLS change detection values was 2.2 mm over the biennial comparison, and thus was in the same order of magnitude as the detected change rates. However, the huge numbers of TLS measurements permit a stable general impression of surface changes, assuming their measurement errors are not spatially correlated (Beer et al., in review). Mean erosion rates of 1 mm/a near the bed gradually decreased to 0.5 mm/a at 0.8 m height (Figure 2 F). Between heights of 1.0 m and 2.0 m erosion rates were more or less constant at 0.5 mm/a, and at higher elevations they quickly approached zero (at 2.7 m, not shown in Figure 2 G). This bedrock erosion pattern reflects the distribution of flushing heights (Figure 2 D), with only brief flushing event peaks exceeding water depths of 2 m, thus delivering few erosive tools to these heights. During the longer time frame of the TLS study, the staff gauge's paint was eroded and re-painted several times. Since successive layers of paint were not located exactly on top of one another, and some locations had more paint during the second scan than during the first, there appears to be an apparent positive surface change (Figure 2 E), which is simply the added thickness of the paint (cf. the blue stripes at heights between 1.0 m and 2.0 m). Likewise, high apparent bedrock erosion rates are indicated at the bottom of the staff gauge (cf. the vertical red stripe pattern below 0.5 m in Figure 2 E), marking regions where paint was present during the first scan but had eroded before the second scan. Thus the calculated erosion rates in Figure 2 E reflect the erosion of both the paint and the bedrock. These distortions of the TLS-based erosion patterns provide a further proof-of-concept of the erosion painting technique, by showing that removal of the paint corresponds to detectable rates of surface erosion. However, they do not distort the general pattern in the erosion profile (Figure 2 F), since that profile is binned over the entire width of the analysed site (cf. Figure 2 E), and thus the influence of the paint is minimized.

At the right gorge wall, both stripes R1 and R2 were eroded in only a restricted band situated more than 15 cm above the bed (Figure 3 for R2). This observation can be explained by the sediment bed aggrading up to this level during flushings, and thus shielding the lower levels of the wall from paint erosion (i.e. the sediment cover effect). The paint was eroded only near the top of this temporary cover in the restricted zone where moving sediment grains (tools) were most abundant (see Turowski et al., 2008). Above and below this restricted zone of complete erosion, only small patchy areas of paint were removed (indicated as "slight erosion" zones; cf. Figure 3 B and C). This patchy erosion can be attributed to selective abrasion of the paint by less frequent sediment impacts than in the zones of full paint erosion. The lower patch of minimal erosion implies that the bed aggraded rapidly at the beginning of flushing events, and degraded rapidly at their ends, leaving little time for paint abrasion. The upper patch of minimal erosion implies rapidly decreasing sediment concentrations in the water column above the temporarily raised sediment bed, and thus generally low sediment transport rates at this location.

Notably, the erosion pattern on the right gorge wall could be detected repeatedly (cf. the three date periods in Figure 3), and the zone of focused erosion on the wall occurred at a consistent height. This suggests that there were only minor fluctuations of bed height and sediment transport on the right side of the gorge, despite differences in flushing durations, in flushing heights (cf. Figure 3 C), and probably also in sediment concentrations and grain sizes. The paint erosion pattern on the right gorge wall (Figure 3 B and C) was not visible in the TLS bedrock change detection study (Beer et al., in review, the right wall is not shown in). Also, the right wall appeared very smooth and did not show any visual evidence of increased abrasion in the zone of complete paint erosion, consistent with low transport rates in this location (as inferred in the previous paragraph). These observations suggest that bedrock erosion rates were too slow to be detected by the TLS surveys, despite visually obvious removal of the (much more erodible) paint. Thus, the erosion painting method may be able to qualitatively detect variations in erosion rates, even when these rates are too low to be measured quantitatively with more sophisticated techniques.

The erosion patterns of the painted surfaces in Figure 4 illustrate how erosion depends on surface orientation and exposure to impacting particles (the tools effect; Beer et al., in review), and on the spatial erosivity of the sediment-laden flow. Zones of focused bedrock erosion, visually inferred from both impact marks and crestlines on the boulder, were confirmed by the distribution of paint erosion: the heavily impacted surfaces were paint-free, and the crestlines formed sharp boundaries delimiting impact-free surfaces that were still paint-covered (Figure 4 A - F, cf. Wilson et al., 2013; Wilson and Lave, 2014). The paint-free upstream face of the slab (Figure 4 H) reflected abundant sediment impacts on this in-stream obstacle (cf. Beer et al., in review). Indeed, most of the few upstream-facing white areas visible in Figure 4 H appear white due to quartz inclusions rather than paint. However, the painted parts of the slab that faced laterally, upward, and downstream (cf. the inset in Figure 4 H) were protected from sediment impacts due to the diversion of sediment tools by

the slab (Beer et al., in review). This indicates the crucial role of streambed topography in guiding streamflow and sediment flux (Johnson and Whipple, 2007, 2010; Cook et al., 2014; Fuller et al., 2016; Beer et al., in review).

A comparative view on the erosion patterns of all the painted stripes on the opposite bedrock walls (Figure 5 A left panel for the period of 06 June 2014 to 09 July 2014; cf. Figure 1 D) revealed

strong cross-sectional differences in the relative importance of the sediment tools and cover effects. Flushed discharge through the gorge carries substantial volumes of sediment that has previously accumulated in the upstream sediment retention basin (Beer et al., in review). Since both the staff gauge and stripe L1 on the left wall were mostly eroded up to 0.7 m above the bed (at least for surfaces facing upstream; Figure 5 B left panel), erosive tools likely abraded the whole left wall with

diminishing intensity with height above the bed (Figure 5 B right panel). In contrast, on the right gorge wall (Figure 5 C left panel), both stripes R1 and R2 showed a very restricted band of erosion (cf. Figure 3), suggesting that here the streambed aggraded up to the same level through multiple flushing events, with only a narrow erosion zone on top of it (Figure 5 C right panel).

Together, these interpretations indicate a strong difference in sediment transport concentration

across the gorge (Figure 5 A right panel): high-velocity transport of large volumes of sediment on the left side, and slower transport of smaller volumes of sediment on the right side, where the sediment bed is elevated due to lower transport capacity. This large difference in sediment transport across a channel width of only 5 m would not have been predicted from the straight channel geometry, from the flat channel bed cross-section at low flows (Figure 5 A left panel), nor from the reasonably

homogeneous water surface across the gorge during flushing events, as observed by eye, in videos, and in pictures (cf. Figure 1 B). The driving mechanism of this laterally focused sediment transport was probably the coarse boulder bed of the channel upstream of the gorge (Figure 1 B) that likely deflected the sediment flow. Directly upstream of the inspected wall section (to the left of Figure 1 C), there are rock blocks of 2 m size in the streambed that leave a passage on the gorge's left side. This

passage may channelize the sediment flow even when these blocks are submerged by the flushing water. Further, secondary currents due to turbulence induced by the boulders are also likely to have influenced the sediment distribution (Venditti et al., 2014). We do not have direct measurements of the spatial sediment transport distribution during the flushings, but the erosion painting technique was able to document the crucial influence of sediment routing in setting local erosion rates.

**4.3  Potential future applications of erosion painting**

Our results demonstrate that erosion painting is a straightforward method for (i) visualizing the spatial distribution of bedrock erosion (i.e. variations with position and orientation), for (ii) inferring the spatial distribution of sediment transport (i.e. the sediment tools and cover effects), and for (iii) localising the transient elevation of the sedimentary streambed under some circumstances. Qualitative

erosion patterns observable in the eroded paint generally coincided with the quantitative bedrock

erosion analysis of Beer et al. (in review), consistently showing that erosion rates of local bedrock surfaces depend on their position in the streambed and their spatial exposure to the impact of erosive tools.

Local erosion rates depend on both the erodibility of the surface and the erosivity of the sediment-laden flow that abrades it. A general challenge in surface erosion studies is that it is difficult to know whether spatial variations in erosion rates are driven by variations in erodibility of the surface or erosivity of the flow. Erosion painting provides an artificial surface (the paint) that has a relatively uniform erodibility, and thus patterns of paint erosion should mostly reflect variations in the erosivity of the streamflow and its entrained sediment. A further step would be to standardize the painting technique to a specified paint volume per unit area, thus better constraining the thickness (and therefore erodibility) of the paint layer. Laboratory tests (e.g., using the erosion mills of Sklar and Dietrich, 2001) could be used to explore the erodibility of different paints, the influence of applied paint thickness, and paint adhesion on different bedrock lithologies. Also, the erosivity of the flow in the abrasion mill (i.e. its ability to erode the paint) could be studied by changing the mill's flow velocity or sediment loading. This analysis could serve as background for more semi-quantitative studies of natural flow erosivities. Further, the choice of a particular paint (with known erodibility) would allow one to specify the threshold above which streamflow erosivity is detectable. Applying a series of layers of this paint, each with a different colour, would permit better quantitative constraints on erosion. Alternatively, one could apply a stack of paint layers with different colours and erodibilities (with each successive layer more erodible than the one below it) to handle a wide range of erosivities.

The simplicity of the erosion painting technique could lead to wide-ranging applications in geomorphology. Examples of advanced applications for field sites like the studied gorge would be (i) to more frequently check eroded paint patterns (e.g., after every erosive event) to find thresholds of paint erosion for constraining streamflow erosivity, (ii) to repeatedly paint entire walls, beds or cross-sections to study the spatial variations in streamflow erosivity due to varying sediment concentrations, or (iii) to paint below the sediment bed or below the on-site water surface to determine how the sediment bed varies during flushings and whether erosion also occurs below the level of the dry bed.

Erosion painting should be applicable to topics and settings well beyond the framework of our study. The relative erodibility of paint by suspended sediment and bedload could be tested in the laboratory, e.g., in experiments similar to those of Attal et al. (2006), Scheingross et al. (2014), or Wilson and Lave (2014). Erosion painting could be used to more rigorously verify the generality of the observation that abrasion by bedload is dominant on stoss surfaces of bedrock, as seen here (cf. Whipple et al., 2000; Beer et al., in review), whereas abrasion by suspended load is more important on lee surfaces (Wilson et al., 2013). The interactions of streambed morphology and sediment routing could also be assessed (cf. Finnegan et al., 2007; Johnson and Whipple, 2007, 2010). Even

erosion depths in (coarse) alluvium could be studied, if a suitable paint is infiltrated into the bed, complementing techniques like scour chains (Laronne et al., 1994; Liebault and Laronne, 2008) or

injection of coloured sand profiles.

Besides application in fluvial environments, erosion painting could also be used to visualize spatial distributions of erosion by ice (e.g., Herman et al., 2015) and wind (e.g., Perkins et al., 2015). Depending on the study topic, erosion painting could be accompanied by, or deliver additional qualitative supporting information for, quantitative surveys of surface change using more sophisticated

instrumentation (e.g., TLS surveys, erosion sensors). For example, erosion painting could be used in pilot studies to provide a qualitative spatial view of local processes (e.g., for planning purposes). It could also enable straightforward comparisons between different sites used for longer-term monitoring, or provide spatial verification for local quantitative studies both in the field and in the laboratory.

*Acknowledgements.* The authors want to thank Rafael Bienz, Jean-Pierre Bloem, Lorenzo Campana, Daniela

Cervenka, Simon Etter, Kristen Cook, Mattia Sieber, Alexander Stahel and Carlos Wyss for helping with painting of bedrock surfaces over the years. We are very thankful to Grande Dixence SA for providing logistic support and discharge data for the Gornera study site. Comments by Joel Johnson, Theodore Fuller and an anonymous reviewer greatly improved this paper. This study was supported by SNF grant 200021 132163/1.

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

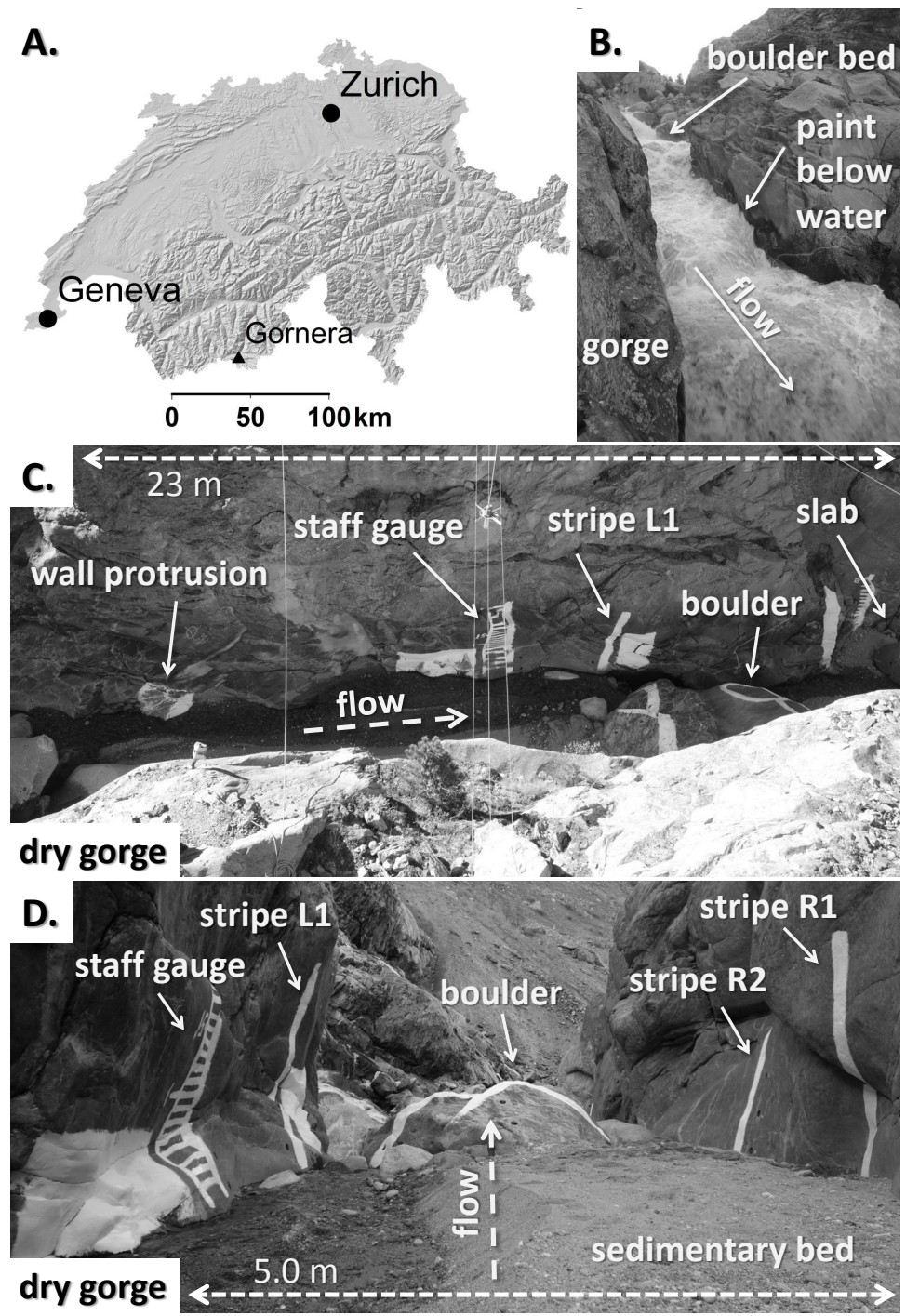

**Figure 1.** The demonstration field site for bedrock erosion painting: (A) The location of the Gornera proglacial stream, Switzerland, (B) lateral view of the bedrock gorge reach during flushing of the sediment retention basin upstream, (C) top view of the gorge reach during dry conditions, showing some eroded painted surfaces on the left wall and in the streambed, and (D) downstream view in gorge reach under dry conditions, showing some of the eroded painted surfaces. Only the paint areas that are indicated and named are used for analysis here.

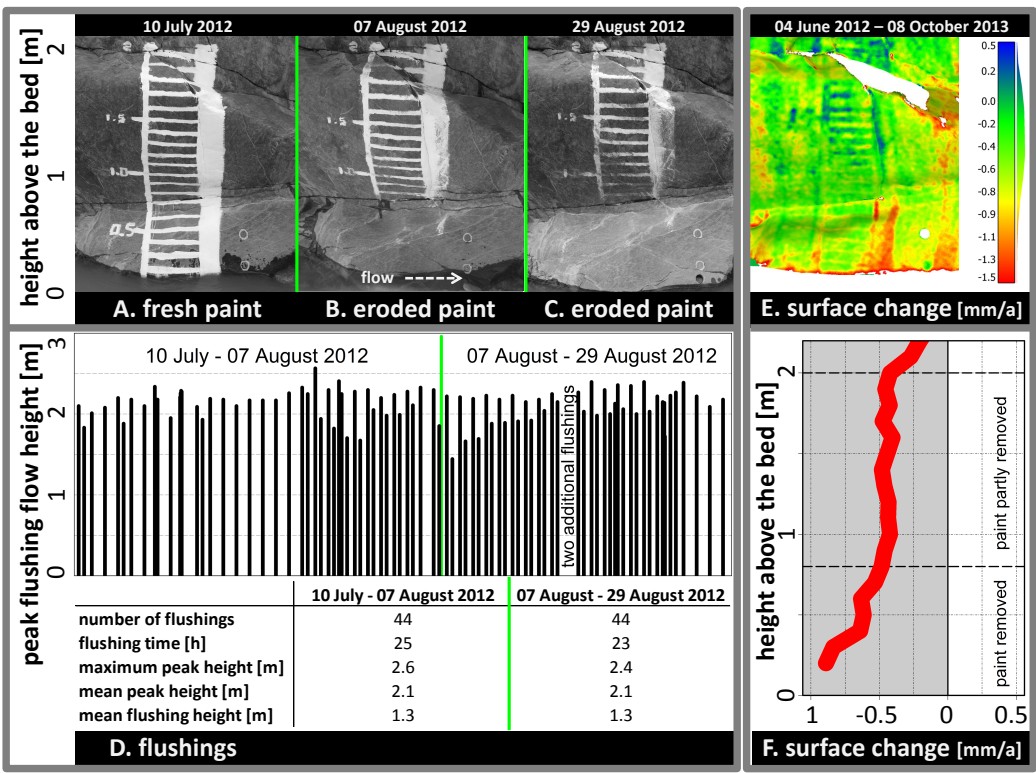

**Figure 2.** The erosion pattern on the painted staff gauge on the left gorge wall (cf. Figure 1 C and D) indicated a region above the streambed in which the sediment tools effect leads to accelerated lateral erosion: (A) the freshly painted staff gauge, (B) the eroded staff gauge after a period of 44 flushing events, (C) the eroded staff gauge (not re-painted on 07 August 2012) after an additional period of 44 flushing events, (D) time series of the flushing event peak flow heights for both periods, (E) mean at-a-point bedrock change detection values (more than 2 million points) from repeated terrestrial laser scanning over the two years 2012 - 2013 (the timespan 04 June 2012 to 08 October 2013 includes nearly all flushing events in these years; for data and calculation see Beer et al., in review), and (F) variation in bedrock erosion rates with height over the streambed. For two flushings of the second painting period, no flushing height data exist (see the data gap in (D)), but the flushing discharge was comparable to the adjacent flushings. Paint erosion and additional paint coating at different positions in between the scanning dates led to some of the extreme change detection values in (E) that reflect the painted gauge pattern (see the text for details). The vertical erosion profile in (F) is based on the mean values of horizontally binned at-a-point erosion rates given in (F), with a bin height of 0.1 m. This profile is only slightly distorted by the erroneous extreme change values from paint erosion (see (E)), due to the huge number of TLS measurements included. The grey background areas in (F) symbolize the region of the bedrock wall, with the change value of 0 mm/a defining its original surface, and erosion penetrating into it. Note the different y-axes of the individual figures.

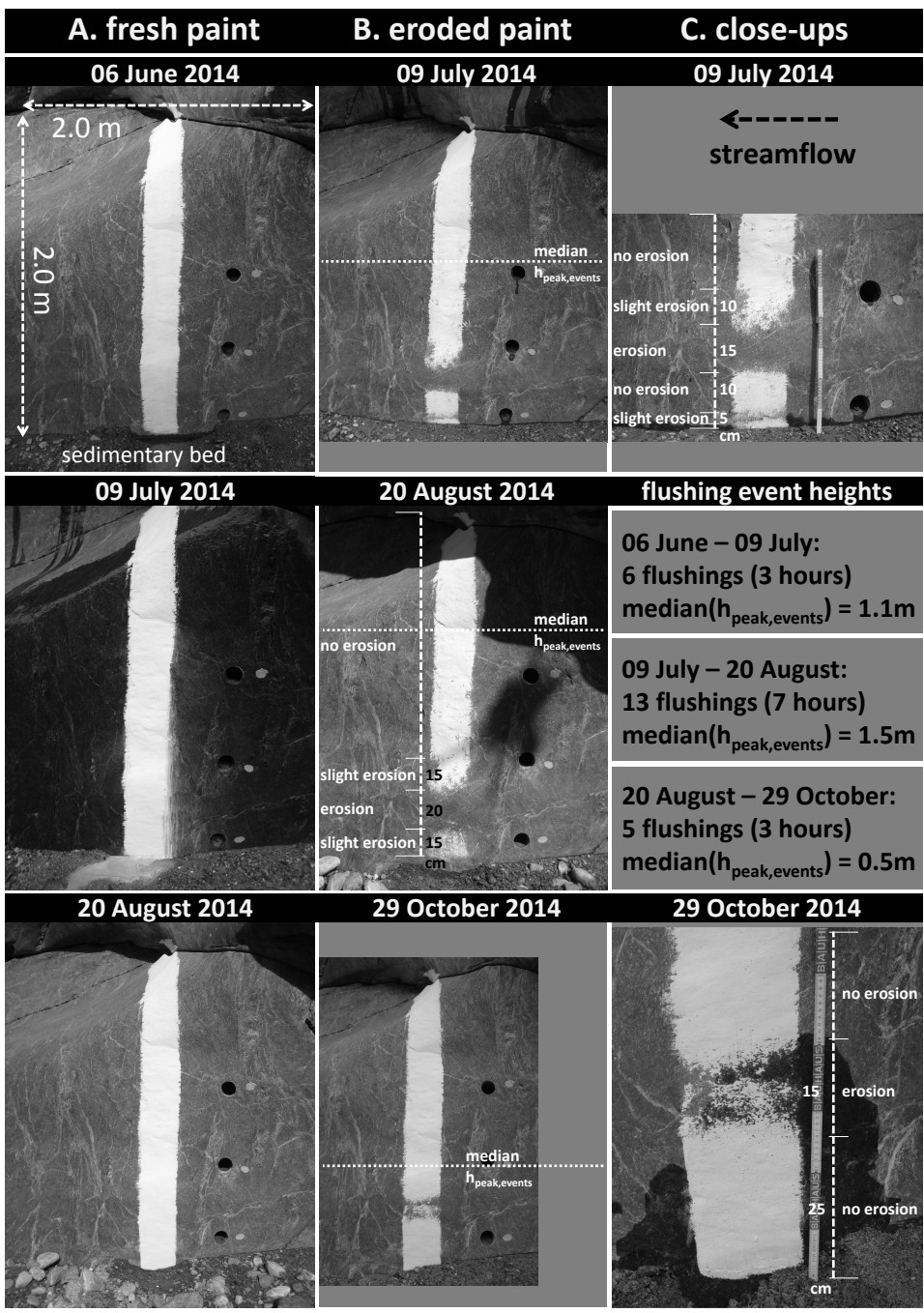

**Figure 3.** Painted stripe R2 on the right gorge wall (cf. Figure 1 D), indicating a zone of complete paint erosion at ~15 - 40 cm above the streambed, suggesting a temporary sediment cover effect due to bed aggregation during flushings and a constrained sediment tools effect on top, causing lateral erosion: Column (A) shows stripe R2 freshly painted on three dates, column (B) shows the same stripe after 4-10 weeks of flushing events, and column (C) shows close-up views of the erosion zone along with information on the flushing events for each of the three time periods. The dotted lines in column (B) locate the median of peak flushing heights per period (median $h_{peak,events}$)

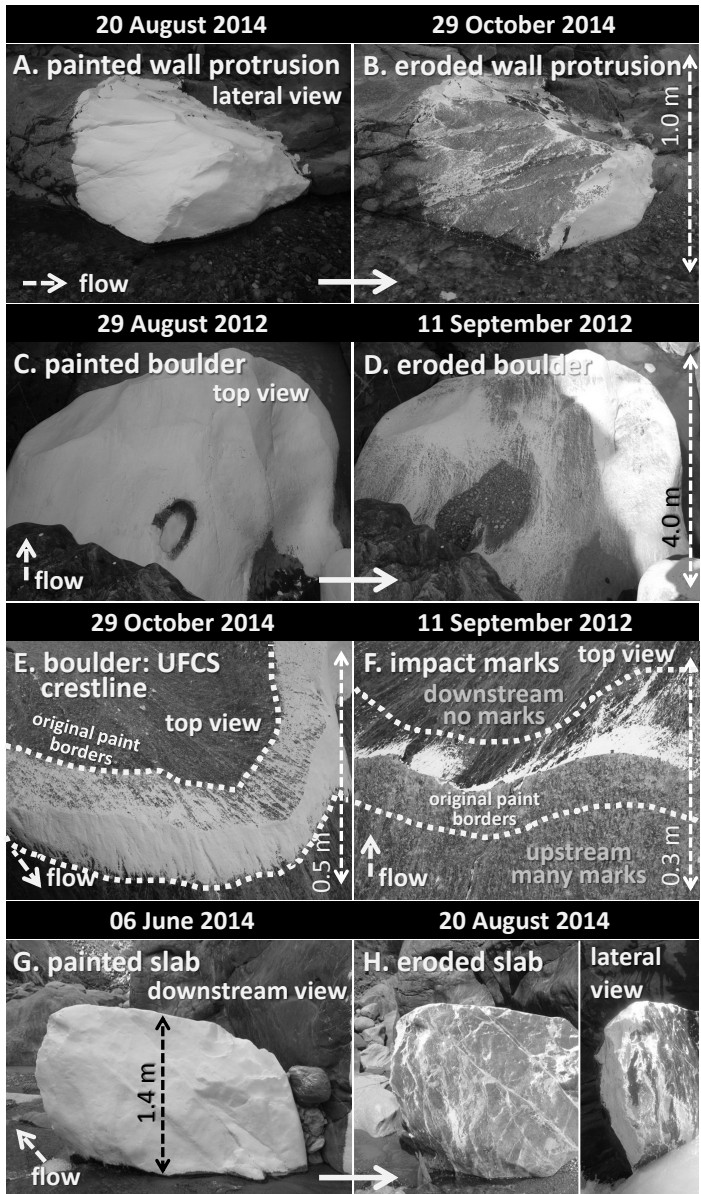

**Figure 4.** Patterns of eroded paint at several sites in the gorge (cf. Figure 1 C and D), illustrating how erosion depends on local surface orientation: (A) lateral view of the painted protruding wall section, (B) eroded paint on that wall section, (C) top view of the painted boulder, (D) eroded paint on that boulder, (E) top view of a boulder crestline that was previously painted on both sides and now only shows erosion on its upstream-facing side, (F) close-up view of a previously painted boulder crestline like in (E), demonstrating sediment impact marks on the upstream side and a lack of impact marks on the downstream side, (G) downstream view of the painted slab, and (H) the eroded paint on the upstream-facing side of that slab, with an additional lateral view of the painted margins of the slab facing upward, downward and laterally (inset on the right). Note the original borders of painting indicated in (E) and (F) by the dotted lines. Bedrock colour differences in (F) are due to the abundant impact marks upstream of the crestline.

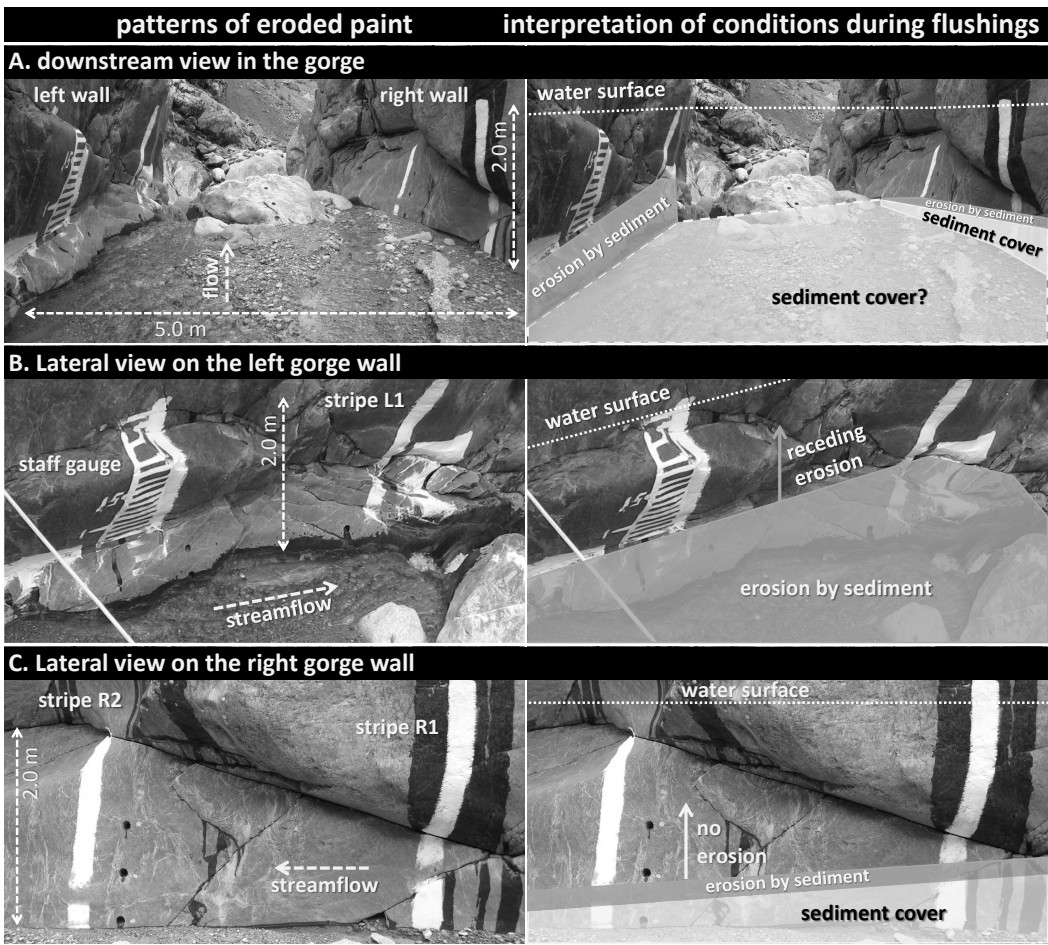

**Figure 5.** Erosion patterns on the painted stripes in the gorge (left column) reflect likely cross-stream variations in the sediment tools and cover effects during flushings (as indicated by intepretive diagrams in the right column): (A) downstream view into the gorge with four painted stripes visible (cf. Figure 1 D), (B) lateral view of the left bedrock wall, and (C) lateral view of the right bedrock wall. All pictures show paint erosion over the period of 06 June 2014 to 09 July 2014. The dark areas in (C) are wet rock sections due to seepage from above.