# Peer review of "Graffiti for science - Erosion painting reveals spatially variable erosivity of sediment-laden flows"

_Earth Surface Dynamics, 2016_

## Referee Comment (RC1) · Anonymous Referee #1 · 28 May 2016

In the submitted manuscript Beer et al propose a new method for documenting spatial distributions in bedrock erosion via painting of bedrock surfaces. The authors suggest that paint is eroded by impacting sediment, such that areas of eroded paint correspond to areas of bedrock erosion, and compare their observations of eroded paint to repeat terrestrial laser scans.

To my knowledge, this is a new technique which has not been previously documented, and the authors show compelling results where the spatial patterns of eroded paint can be used to infer both variations in sediment impacts across a channel, as well as temporary aggradation of the sediment bed which can occur during floods. Erosion monitoring via painting appears to be a useful and easily applicable tool which can

be of use in a variety of erosion studies, and as such, this manuscript deserves to be published. However, I have a few general comments which should be considered before published the final version.

Major comments

1. The repeat photographs showing eroded paint do an excellent job of documenting erosion of paint via sediment impacts. However, I found the comparison with the TLS scans quite useful for making the connection that erosion of paint tracks with actual bedrock erosion. As this is intended to be a proof-of-concept methods paper, I think the paper could benefit from more comparisons between the TLS scans and photographs. For example, it could be useful to show some type of quantitative comparison between the eroded paint and documented bedrock erosion from the TLS scan (if the photos can be mapped over the TLS scans, then one could, for example, directly compare the areas of eroded paint with bedrock erosion from the TLS scans). For the paired TLS scan and eroded paint pair presented, it would be useful to show a TLS pair which corresponds to the photograph dates for painting (if such a TLS pair exists). Additionally, showing other TLS pairs which correspond to the eroded paint at different locations in the gorge would be worthwhile.

2. This appears to be a companion paper to a manuscript the authors have in review at JGR – Earth Surface. I have not seen this other manuscript in review, but from the citations listed here, this other paper appears to present more of the science and implications associated with documenting spatially-variable erosion, while this manuscript is focused more on methods. To that extent, there's large portions of the discussion section (e.g., section 4.2) as well as other parts of the paper that address some of the process implications and general science questions which may be more appropriate for the companion paper. Removing such sections from this paper would shorten the manuscript length and help to keep the focus on presenting a new method rather than discussion of science. Minor comments

1. The authors frequently use the terminology which is either not explicitly defined or can have an ambiguous meaning. For example, the terms "tools effect" and "cover effect" should be defined since they won't be obvious to all readers. Furthermore, I think it is more straightforward to simply describe the actual processes going on rather than using terminology, so in many places use of the term "tools effect" could be replaced, for example, with "sediment impacts", and similarly "cover effect" could be replaced with something like "shielding of bedrock by deposited sediment."

Often times erosion is mentioned, but it is unclear if this is meant to be erosion of paint or erosion of bedrock. Also, I think the authors use the term "erodibility" to refer to erosion of a material (i.e., bedrock erosion or erosion of paint), while "erosivity" refers to the ability of a sediment-laden flow to erode material, but I don't think these terms were explicitly defined.

2. The writing in the paper is acceptable and the paper is readable, but there are many examples throughout the paper where the writing can be made more concise and superfluous information could be removed to reduce the overall manuscript length. I've indicated some of these in the technical comments below, but there are more throughout the paper.

3. The documentation of aggradation during floods is cool and perhaps worth also highlighting in the abstract of the paper and emphasizing as an important benefit of this technique.

Line-by-line comments (no response needed)

Throughout the manuscript and the figures all dates should be listed in international format, i.e., 29.08.2012 should become 29 August 2012.

L5 and L8 – Suggest removing the terms tools and cover effects from the abstract, and instead replacing with sediment impacts and deposited sediment since these are

non-technical terms, and tools/cover effect should be first defined.

L13 – The 'several interacting processes' are not mentioned, which makes this sentence awkward.

L14 – Suggest 'hydraulic shear detachment of bedrock grains' and 'plucking of bedrock blocks' or something similar for clarity.

L21 – Suggest 'downstream directed erosion of bedrock which protrudes into the flow' (I was confused when I first read this sentence, I don't think I've ever seen the term 'channel clearance' before).

L5-12 – This paragraph is introductory material, not methods material (at least in my mind...).

L6-8 – Suggest removing "Besides colouring such tracers" and combining the two sentences for clarity.

L15 – "Defined vantage points" is maybe better as "repeatable vantage points"

L17 – Are "retrievable features" benchmarks? If so, suggest a change of terminology

L19 – This is a claim that will be later documented in the paper, but it is written here as fact. For the methods, I suggest writing something like "We suggest that the spatial pattern..."

L24 – TLS should appear in parentheses, e.g., (TLS).

L1-3 – Suggest, "We tested the ability of paint to record spatial variations in bedrock erosion by comparing photos of worn paint to quantitative erosion analyses based on repeat high-resoltion TLS surveys.

L5 – Suggest starting with at least one or two general sentences describing the erosion

that occurred in the gorge rather than jumping right into the specific details.

L8 – Here and also on L11-12 the writing is a bit ambiguous. I think you mean the highest flushing height recorded over the entire period was 2.6 m, the mean height averaged over all the events (including the time discharge was building up and waning down) was 1.3 m. Also, for the purposes of this methods, proof-of-concept paper, I'm not sure all this information is needed and some of it could be summarized in a table (and I imagine a lot of it is reported in the JGR paper), so some of it could be removed here to shorten the manuscript.

L10 – Can you provide some explanation why it is likely that there were three more flushings? Did the power company report three flushings but no equipment was installed to measure the water flow at the time?

L23 – The word choice "interesting" here is not a very useful word as it is subjective and does not describe any of the actual patterns.

L26 – Is "UFCS-form evolution" a term that is commonly used in the literature? This seems like imprecise English to me, maybe try something like "The eroded paint on the boulder showed patterns that are typical for forming and upstream facing convex surfaces..."

L1 – Suggest sediment flow should be sediment impacts.

L7 – I would add water dissolution in addition to shear detachment.

L10-15 – I see the erosion painting method as a quantitative measure of the spatial distribution of sediment impacts, so I think this is something you could include if you wanted to mention a quantitative ability of the method. Agreed that erosion measurements from the method are qualitative.

L4-5 (and elsewhere) – I suggest reporting erosion rates in either mm/yr (instead of mm per 2 years, which is quite uncommon), and these should be instantaneous erosion rates based on the total time of flushing. Because the gorge's hydraulic regime is not natural, it makes more sense to me to normalize erosion rates by the total flushing time, as the flushing time can vary from year to year.

L24-25 – I found it odd that shear detachment of paint by bubbles was mentioned here and not discussed. If this is to be brought up, the authors should discuss to the extent possible whether or not this mechanism led to erosion of the paint. My interpretation is that there are areas which were inundated with water but do not show erosion, such that the water alone is not able to erode the paint.

L32 – This is the only time in the manuscript where the possibility of varying sediment size is mentioned. If there is some data on this, it could be useful to mention earlier in the manuscript (this doesn't need to be in a lot of detail, but when describing the different flushing heights, one could also mention how the size of sediment supplied to the gorge could have changed).

L33-35 – This is a nice example of where the authors could show more TLS data as an example of when TLS measures no erosion but painting shows sediment impacts.

L4-5 "who showed" should be "which shows"

L7 – The phrase "From a more local perspective" doesn't really make sense to me in this context. Also, here is an example of where erodibility and erosivity need to be explicitly defined.

L11-13 – Abrasion mills can also be used to explore variations in "erosivity" by changing the flow velocity, the sediment size, or the sediment load, so I find this comparison a bit unfair or perhaps misleading.

L26 – I think the study here already demonstrates a test of relative erodibility of paint

ESurfD
by water, doesn't it?

L32 – Hasn't erosion of alluvium already been studied using paint in the Dietrich et al (2005) and Surian et al (2009) refs already cited? Or did you have something else in mind (if so, please elaborate).

Figure 2

- Again, units of mm/2year seem inconvenient to me. Suggest changing as mentioned above.

- 2E – Instead of having an x-axis on the bar chart, I suggest simply listing the date interval over which each bar corresponds to.

- 2D and E – Are these peak flushing heights?

- In the caption for (F) please list the exact date range rather than over "two years 2012-2013"

- 2G - Please report width of bin measurements are averaged over.

Figure 3

- The numbers indicating (I think) height in cm of different features along the painted stripes are not clear. The unit should be given and I also suggest adding a double-pointed arrow to make it totally clear over what area the number applies to.

- Showing the average or peak water surface height on these photos would be good if possible (if water surface is above the top of the photo, that could be noted in the caption).

- Use of "first period", "second period", etc is slightly ambiguous, instead I suggest listing the actual date ranges. Also is there not a close up photo for the second row?

Figure 4

- I had a hard time mapping photos E and F to their location in the channel on Figure

1, are they on there? Maybe this could be made more clear.

- Do impact marks refer to impacts of grains eroding paint, or groves, scours, etc. on the actual bedrock surface from impacting sediment? Perhaps showing a close-up of these impact marks would be helpful.

Figure 5

- I would change "assumed conditions" to "interpretation of conditions"

---

## Referee Comment (RC2) · J. Johnson (Referee) · 19 Jun 2016

"Graffiti for science" by Beer et al. presents an interesting proof-of-concept study that nicely illustrates the utility of using paint to indicate spatial patterns of surface erosion or sediment transport. I recommend publication with minor revisions to clarify just a few points. While I think the idea of using paint to constrain erosion patterns is fairly straightforward, the case study and field site is sufficiently novel and detailed to warrant publication. I believe this work will inspire others to use the technique.

Page 1 line 9: I appreciate that the authors do not oversell their technique, but in some ways my initial reaction to this line is that "qualitative" sells their method a little short. True that they cannot quantitatively measure erosion rates with the technique (or at

least not with much accuracy), but using paint can quantitatively give spatial pattern of surface impacts. I can live with calling the method qualitative, but would encourage selling it as potentially a way to quantify spatial patterns. Same comment would apply to some points in the discussion.

Pg1line14: Not sure its worth separating out hydraulic shear detachment and plucking as separate processes. I could be wrong (didn't go back and check), but I don't remember any of the papers they reference for this point emphasizing hydraulic shear detachment in rock separate from plucking.

Pg1 line 24: The authors could consider referencing Johnson, Whipple, Sklar, GSA Bulletin 2010, "Contrasting bedrock incision rates from snowmelt and flash floods in the Henry Mountains, Utah", which monitored bedrock incision in a natural channel (albeit in a modified reach), and focused on spatial patterns of incision in relation to sediment transport and accumulation as well as hydrographs.

Pg2 line 14: Maybe a little more detail on the paint. Was it housepaint? Latex based? Oil based? There are a great many types of paint with different properties that could be described as environmentally safe paint for outdoor use.

Pg3 line10: of 42 rather than 42 of?

Pg3line11: say a bit more about the data gap—which data set? Flow depths? Was it a sensor failure? Does the power company not have records of releases?

Pg3 line27: describe "vertical erosion on planar surfaces" a bit more; I don't think this is a specific enough description. Lots of surfaces oriented differently relative to flow are probably planar but don't have vertical erosion in this study.

Pg4 lines19-20: I realize references are given for erosion and erosivity, but I think it worth clarifying a bit more what is meant here. i.e., say something like the amount (length) of vertical erosion.

Pg4 line 23: Probably don't need to point out that permission could be needed to do

graffiti. . .

Pg5 line9-17: I think briefly mention this explanation in the caption of figure2, and/or in the text where the figure is referenced, or at least say something like "relation between the change detection and painted areas is further explained in the discussion section". When I looked at figure 2 I tried to figure out what was going on with the patterns of erosion on the surface, and was perplexed until getting to this paragraph.

Pg5 lines 24-27: So do the authors think these issues—air bubbles in the paint, and painting on wet rock surfaces—affected their measurements? It's a little unclear how much these factors may have influenced their results. Is it conjecture, or based on hindsight from their results?

Pg7 lines 9-12: I'm not sure how well this would necessarily work, because how well paint adheres to a surface is generally pretty sensitive to the surface characteristics. I would think that different rock types or compositions would typically have different "paintabilities". Its kind of hard to get paint to stick well to quartz, for example. In any case this effect should be acknowledged.

Figure 2: In 2f and 2g, does "mm/2a" mean millimeters per two years? So is this not the amount of erosion measured between 4/6/2012 and 8/10/2013 (1 year and 4 months or so), but that amount of erosion, normalized up to two years? Please clarify what the 2 years means in this case.

I realize the Lidar-based change detection measurements are not the main focus of the present work, but on the figure or in the main text I think some mention of uncertainty of these measurements is needed. Both uncertainty of the individual scans themselves, and also uncertainty in the differences.

In 2e, it seems confusing to have the box and whisker plot rectangles not be centered over the time intervals that I think they're supposed to represent.

In figure 5c, I presume the dark vertical lines on either side of the white painted area is

the rock being wet? Clarify in caption what the dark areas are. Conceivably could be intrusions or something.

Joel Johnson

---

## Referee Comment (RC3) · T. Fuller (Referee) · 20 Jun 2016

This is a welcome addition to the literature on erosion in bedrock channels. One of the primary contributions of the paper is the presentation of field-based evidence that supports several hypotheses of bedrock channel erosion. The field observations presented here support at least three existing hypotheses on bedrock erosion processes: 1) bedrock erosion rates are highly variable depending on the in-channel orientation of bedrock surfaces; 2) lateral bedrock erosion via tools is focused at the base of the channel wall; 3) suspended sediment is capable of bedrock erosion.

Field-based evidence of a substantial cross-channel gradient in sediment concentration during high, erosive channel flows is a valuable contribution to the literature. As such,

[Figure]

I would like to see this explored in a little more detail (to the extent possible) perhaps by providing basic hydraulic conditions within and upstream of the gorge that might be driving this apparent concentration gradient.

As a proof-of-concept study, I would have liked to see a more detailed discussion of the methodology. The authors briefly mention some of the potential pitfalls of the method in the discussion section but do not really address how these pitfalls can be avoided. Maybe the authors have indeed provided the necessary level of detail for a proof-of-concept study, perhaps the editor has a better feeling for the detailed required for such a study.

In light of the comment above, it might be prudent to focus a little less on the proof-of-concept idea, particularly in the abstract and introduction sections, and more on the qualitative field-based evidence which lends support to several existing hypotheses. Again, I think perhaps the most exciting thing here is that you have FIELD-BASED evidence of patterns of bedrock erosion.

More detail of the field site would help the reader put the findings in context. Estimate of hydraulic conditions, grain size and mobility, primary lithology of channel boundaries. . .

Line-by-line comments submitted as a supplemental document using the Adobe comment tool within the manuscript.

Please also note the supplement to this comment:
http://www.earth-surf-dynam-discuss.net/esurf-2016-27/esurf-2016-27-RC3-supplement.pdf

[Figure]

**Supplement:**

[revised manuscript text omitted]

---

## Author Comment (AC1) · 19 Oct 2016

**Reply to the comments of the three referees**

We are grateful to Joel Johnson, Theodore Fuller, and the anonymous referee for their careful reading of our manuscript, their very encouraging comments, and their useful hints! In the following, we will reply to their major and minor comments by first quoting their words (in normal font) and then providing our response (in blue and italic font).

Alexander Beer, James Kirchner, and Jens Turowski
* * *
**A. Reply to the comments of Anonymous Referee #1**

In the submitted manuscript Beer et al propose a new method for documenting spatial distributions in bedrock erosion via painting of bedrock surfaces. The authors suggest that paint is eroded by impacting sediment, such that areas of eroded paint correspond to areas of bedrock erosion, and compare their observations of eroded paint to repeat terrestrial laser scans.

To my knowledge, this is a new technique which has not been previously documented, and the authors show compelling results where the spatial patterns of eroded paint can be used to infer both variations in sediment impacts across a channel, as well as temporary aggradation of the sediment bed which can occur during floods. Erosion monitoring via painting appears to be a useful and easily applicable tool which can be of use in a variety of erosion studies, and as such, this manuscript deserves to be published. However, I have a few general comments which should be considered before published the final version.

> *Thank you for your kind evaluation of our work.*

**Major comments**

1. The repeat photographs showing eroded paint do an excellent job of documenting erosion of paint via sediment impacts. However, I found the comparison with the TLS scans quite useful for making the connection that erosion of paint tracks with actual bedrock erosion. As this is intended to be a proof-of-concept methods paper, I think the paper could benefit from more comparisons between the TLS scans and photographs. For example, it could be useful to show some type of quantitative comparison between the eroded paint and documented bedrock erosion from the TLS scan (if the photos can be mapped over the TLS scans, then one could, for example, directly compare the areas of eroded paint with bedrock erosion from the TLS scans). For the paired TLS scan and eroded paint pair presented, it would be useful to show a TLS pair which corresponds to the photograph dates for painting (if such a TLS pair exists). Additionally, showing other TLS pairs which correspond to the eroded paint at different locations in the gorge would be worthwhile.

*We are happy to hear that the reviewer agrees that the photo-comparison method is useful for easily demonstrating spatial erosion distribution. Unfortunately, no further direct comparative TLS data sets exist (specifically for the same period as the photographs), either because of low bedrock erosion rates, so that no clear pattern was visible (as noted in the text for the right gorge wall), or due to high bedrock erosion rates, so that all the paint was removed (as for the slab, Figure 4H). Direct quantitative comparison of paint erosion with TLS scans is problematic, because the paint is removed by amounts of erosion that are barely detectable by TLS, and, conversely, because erosion that can be easily measured by TLS would have removed the paint many times over.*

*We tried overlaying photos on TLS change detection data (and vice versa), but the resulting images were indecipherable due to the combination of the natural shading of the photos and the artificial shading of the TLS data. We think that the comparison of Figure 2B/C and 2F already shows that erosion painting clearly identifies the zones most affected by surface erosion. We also do not have directly comparable scans and paintings, since we painted much more often than we scanned.*

*Further (as outlined in more detail in the response to the second major comment below), this article on erosion painting is intended to provide both (i) a proof-of-concept study on the method and (ii) a discussion of its applicability, its advantages and its scientific potential, by highlighting specific results from its first application at our field site. Therefore, the focus of this article is not only on showing the comparability of the erosion patterns from TLS-based change detection and erosion painting. To more clearly indicate that this is not solely a proof-of-concept study, we will add to the beginning of the discussion: "In the following, we first assess the erosion painting method based on our proof-of-concept study. We then use this technique to draw inferences about spatial erosion processes at our study site, and discuss potential future applications in the geosciences.".*

2. This appears to be a companion paper to a manuscript the authors have in review at JGR – Earth Surface. I have not seen this other manuscript in review, but from the citations listed here, this other paper appears to present more of the science and implications associated with documenting spatially-variable erosion, while this manuscript is focused more on methods. To that extent, there's large portions of the discussion section (e.g., section 4.2) as well as other parts of the paper that address some of the process implications and general science questions which may be more appropriate for the companion paper. Removing such sections from this paper would shorten the manuscript length and help to keep the focus on presenting a new method rather than discussion of science.

*The present manuscript is not a companion paper to the one under review at JGR-ES, but an independent study with its own measurement technique and scientific results. The JGR paper presents the laser scans and their interpretation and does not mention the erosion*

*painting method at all. We point to the JGR paper for more detailed information on TLS data acquisition, processing and analysis for comparison with erosion painting.*

*In this ESurf paper, we not only address the scientific usefulness of the erosion painting method and outline potential areas of application, but we identify and discuss specific spatial patterns observed with this method in the field: (i) indirect visualisation of the tools effect (by the impact marks in the paint), (ii) consistent erosion patterns over different time periods, and (iii) strong variations in bedload transport across the study gorge. These are results that we could not obtain using the TLS scans. Hence, it would not be appropriate to move the discussion of these new observations and findings (e.g., section 4.2) to the JGR paper.*

**Minor comments**

1. The authors frequently use the terminology which is either not explicitly defined or can have an ambiguous meaning. For example, the terms "tools effect" and "cover effect" should be defined since they won't be obvious to all readers. Furthermore, I think it is more straightforward to simply describe the actual processes going on rather than using terminology, so in many places use of the term "tools effect" could be replaced, for example, with "sediment impacts", and similarly "cover effect" could be replaced with something like "shielding of bedrock by deposited sediment."
Often times erosion is mentioned, but it is unclear if this is meant to be erosion of paint or erosion of bedrock. Also, I think the authors use the term "erodibility" to refer to erosion of a material (i.e., bedrock erosion or erosion of paint), while "erosivity" refers to the ability of a sediment-laden flow to erode material, but I don't think these terms were explicitly defined.

> *Thank you for pointing out these issues! We will clearly define these terms at their first use (tools and cover effects, erodibility and erosivity), and refrain from using them when direct description is more practical.*

2. The writing in the paper is acceptable and the paper is readable, but there are many examples throughout the paper where the writing can be made more concise and superfluous information could be removed to reduce the overall manuscript length. I've indicated some of these in the technical comments below, but there are more throughout the paper.

> *In addition to responding to these technical comments below, we will the whole manuscript again to see where we can remove superfluous wording.*

3. The documentation of aggradation during floods is cool and perhaps worth also highlighting in the abstract of the paper and emphasizing as an important benefit of this technique.

> *We agree and will gladly add a note to the abstract!*

Line-by-line comments (no response needed)

*Thank you very much for the thorough and helpful comments, we will gladly address them! Here are some responses to several of the line-by-line comments.*

L13 – The 'several interacting processes' are not mentioned, which makes this sentence awkward.

*Actually, in the further course of this sentence we name these processes (hydraulic shear detachment, plucking, abrasion, dissolution, and cavitation), but we will divide this sentence to be more clear: "Bedrock erosion in stream channels is driven by several interacting processes, of which the most efficient are hydraulic shear detachment of weak bedrock, plucking of bedrock blocks, and abrasion of small bedrock grains due to sediment impacts. Dissolution and cavitation can also be important contributors to bedrock erosion under specific conditions (Whipple et al., 2000; Sklar and Dietrich, 2004; Lamb et al., 2008; Vachtman and Laronne, 2013).".*

Page 3 L8 – Here and also on L11-12 the writing is a bit ambiguous. I think you mean the highest flushing height recorded over the entire period was 2.6 m, the mean height averaged over all the events (including the time discharge was building up and waning down) was 1.3 m. Also, for the purposes of this methods, proof-of-concept paper, I'm not sure all this information is needed and some of it could be summarized in a table (and I imagine a lot of it is reported in the JGR paper), so some of it could be removed here to shorten the manuscript.

*You are right, these sentences are a little hard to read. We therefore will move these values to a table in Figure 2 and shorten the sentences here, only stating that both flushing periods are comparable. However, as detailed above, this paper includes scientific results and interpretation (based on these numbers), and we therefore need to keep these data.*

Page 4 L10-15 – I see the erosion painting method as a quantitative measure of the spatial distribution of sediment impacts, …

*Thank you for that hint! We will gladly extend this section as follows: "Hence, erosion painting provides a semi-quantitative measure of the spatial distribution of sediment impact intensity, i.e. the erosivity of the streamflow. Assuming that impacting grains that remove the paint also abrade the underlying bedrock (which is reasonable from Figure 2 E, and from the impact marks in Figure 4 F), the erosion painting procedure can be further considered as an indirect measure of bedrock erosion.".*

Page 5 L4-5 (and elsewhere) – I suggest reporting erosion rates in either mm/yr (instead of mm per 2 years, which is quite uncommon), and these should be instantaneous erosion rates based on the total time of flushing. Because the gorge's hydraulic regime is not natural, it makes more sense to me to normalize erosion rates by the total flushing time, as the flushing time can vary from year to year.

*We agree that reporting erosion rates in mm/2a is uncommon and will therefore present these rates in mm/a. We consider the measured erosion rates over the two years as realistic estimates of annual means for the man-made hydraulic regime at the Gornera, since the erosion values of the TLS measurements in Figure 2 F and G date from a period of around 200 erosive flushing events over two years with very different flushing behaviour (frequency, length, volumes etc., as outlined in Beer et al., in review). Hence, we consider it appropriate to report annual mean erosion rates instead of instantaneous values per flushing time.*

*We will mention this in the text: "Quantitative TLS-based spatial bedrock erosion measurements (over two years with more than 200 flushing events of various discharges, length and volumes; see Figure 2 E) confirmed the decrease in the tools effect with height above the bed, as qualitatively inferred from erosion painting.".*

*We doubt that erosion rates scale linearly with either the number of flushing events or the total flushing time, because (for example) flushing twice as often would entail half as much sediment being transported per flushing event (thus the number of events would double and the amount of sediment would remain the same). Likewise flushing twice as long would not change the total volume of sediment transported through the study reach (even though the total flushing time would double).*

Page 5 L24-25 – I found it odd that shear detachment of paint by bubbles was mentioned here and not discussed. If this is to be brought up, the authors should discuss to the extent possible whether or not this mechanism led to erosion of the paint. My interpretation is that there are areas which were inundated with water but do not show erosion, such that the water alone is not able to erode the paint.

*We will move the discussion of potential incorporation of air bubbles in the paint to the subsection "General assessment of the erosion painting technique". Actually, there were no indications of paint erosion patterns influenced by included air bubbles, but we will mention the need to ensure dry and clean conditions during painting to avoid this problem: "The paint should be applied carefully (e.g., avoiding wet and dusty rock, and leaving sufficient time for drying), since incorporated air bubbles or insufficient drying could lead to shear detachment of the paint by flowing water alone, without abrasion of the surface.".*

Page 5 L32 – This is the only time in the manuscript where the possibility of varying sediment size is mentioned. If there is some data on this, it could be useful to mention earlier in the manuscript (this doesn't need to be in a lot of detail, but when describing the different flushing heights, one could also mention how the size of sediment supplied to the gorge could have changed).

*Unfortunately, we do not have data on transported grain sizes, but due to the differences in flushing discharges, lengths, frequencies and volumes there may have been differences in*

*transported grain sizes and sediment concentrations. We will address this in the section on the field site: "Due to the characteristics of the flushing operations (i.e. short, steep hydrographs and evacuation of previously accumulated sediment), the mean transported bedload grain size ($D_{50}$) likely varies considerably during each flushing event and between flushing events. The $D_{50}$ of the natural streambed upstream of the hydropower water intake is ~4 cm at low flows, but it is unknown whether the average sediment load (including high flows) is finer or coarser than this.".*

Page 7 L4-5 "who showed" should be "which shows"

*Actually, both are possible, but we will streamline this sentence as follows: "Qualitative erosion patterns observable in the eroded paint generally coincided with the quantitative bedrock erosion analysis of (Beer et al., in review), consistently showing that erosion rates of local bedrock surfaces depend on their position in the streambed and their spatial exposure to the impact of erosive tools.".*

Page 7 L11-13 – Abrasion mills can also be used to explore variations in "erosivity" by changing the flow velocity, the sediment size, or the sediment load, so I find this comparison a bit unfair or perhaps misleading.

*Oh yes, correct, we did not consider that! Thank you for the hint! We will remove this erroneous notion in the text, but we will gladly build on your comment and mention that abrasion mills could be used to study the influence of streamflow erosivity on paint erosion: "Laboratory tests (e.g., using the erosion mills of Sklar and Dietrich,2001) could be used to explore the erodibility of different paints, the influence of applied paint thickness, and paint adhesion on different bedrock lithologies. Also the erosivity of the flow in the abrasion mill (i.e. its ability to erode the paint) could be studied by changing the mill's flow velocity or sediment loading. This analysis could serve as background for more semi-quantitative studies of natural flow erosivities.".*

Page 7 L32 – Hasn't erosion of alluvium already been studied using paint in the Dietrich et al (2005) and Surian et al (2009) refs already cited? Or did you have something else in mind (if so, please elaborate).

*Thank you for pointing this out! Dietrich et al., 2005 and Surian et al., 2009 studied changes of the upper sediment layer (or armour layer). However, we here mean studying the erosion depth of alluvium. Hence, we write instead: "Even erosion depths in (coarse) alluvium could be studied, if a~suitable paint is infiltrated into the bed, complementing techniques like scour chains (Laronne et al., 1994; Liebault and Laronne, 2008) or injection of coloured sand profiles.".*

**B. Reply to the comments of Joel Johnson**

"Graffiti for science" by Beer et al. presents an interesting proof-of-concept study that nicely illustrates the utility of using paint to indicate spatial patterns of surface erosion or sediment transport. I recommend publication with minor revisions to clarify just a few points. While I think the idea of using paint to constrain erosion patterns is fairly straightforward, the case study and field site is sufficiently novel and detailed to warrant publication. I believe this work will inspire others to use the technique.

*Thank you for your kind recognition of our work!*

**Comments**

Page 1 line 9: I appreciate that the authors do not oversell their technique, but in some ways my initial reaction to this line is that "qualitative" sells their method a little short. True that they cannot quantitatively measure erosion rates with the technique (or at least not with much accuracy), but using paint can quantitatively give spatial pattern of surface impacts. I can live with calling the method qualitative, but would encourage selling it as potentially a way to quantify spatial patterns. Same comment would apply to some points in the discussion.

*Thanks for your encouragement, will gladly take your hint! Actually, the painting technique does not really give a quantitative measure (it would need to be calibrated against independent measurements), but it could serve as a semi-quantitative measure of sediment impact intensities (i.e. showing regions with different partial paint removal). Hence, in the abstract we will state: "The erosion painting method provides a simple technique for mapping sediment impact intensities, and qualitatively observing spatially distributed erosion in bedrock stream reaches.".*

*In the chapter "General assessment of the erosion painting technique" we will discuss this writing: "The transient paint erosion on the higher parts of the staff gauge between Figure 2 A and C, and also the slight erosion zones above and below the zone of complete erosion in Figure 3 B, indicated regions with lower sediment impact frequencies. Hence, erosion painting provides a semi-quantitative measure of the spatial distribution of sediment impact intensity.".*

Pg1line14: Not sure its worth separating out hydraulic shear detachment and plucking as separate processes. I could be wrong (didn't go back and check), but I don't remember any of the papers they reference for this point emphasizing hydraulic shear detachment in rock separate from plucking.

*Correct; we will add another reference here that deals with hydraulic shear detachment as the dominant erosion process (Vachtman and Laronne, Geomorphology 2013: Hydraulic geometry of cohesive channels undergoing base level drop).*

Pg1 line 24: The authors could consider referencing Johnson, Whipple, Sklar, GSA

Bulletin 2010, "Contrasting bedrock incision rates from snowmelt and flash floods in the Henry Mountains, Utah", which monitored bedrock incision in a natural channel (albeit in a modified reach), and focused on spatial patterns of incision in relation to sediment transport and accumulation as well as hydrographs.

*Thanks, we will gladly cite it.*

Pg2 line 14: Maybe a little more detail on the paint. Was it housepaint? Latex based? Oil based? There are a great many types of paint with different properties that could be described as environmentally safe paint for outdoor use.

*We will expand our description here to: "We used environmentally safe and water-insoluble latex-based dispersion paint to cover natural bedrock surfaces ...".*

Pg3 line10: of 42 rather than 42 of?

*Yes ☺, we will change this accordingly.*

Pg3line11: say a bit more about the data gapăˇAˇTwhich data set? Flow depths? Was it a sensor failure? Does the power company not have records of releases?

*Actually, we only have a data gap in our flushing height measurements, but know of two flushing events from the hydropower discharge data (that were similar to the adjacent flushings). We will include the description of the flushings in a table added to Figure 2 to give a better overview and save space.*

Pg3 line27: describe "vertical erosion on planar surfaces" a bit more; I don't think this is a specific enough description. Lots of surfaces oriented differently relative to flow are probably planar but don't have vertical erosion in this study.

*To be more precise we will extend this as follows "… vertical erosion on planar surfaces in line with the streambed …".*

Pg4 lines19-20: I realize references are given for erosion and erosivity, but I think it worth clarifying a bit more what is meant here. i.e., say something like the amount (length) of vertical erosion.

*We will change and shorten this sentence for more clarity: "However, drawing quantitative inferences on erosion rates would require calibration against independent measurements, because the erodibility of the paint and the underlying bedrock will typically differ by large factors (see further discussion below).".*

Pg4 line 23: Probably don't need to point out that permission could be needed to do graffiti …

*We think it is better to mention this, since we would not want readers to think we were advocating indiscriminate use of graffiti.*

Pg5 line9-17: I think briefly mention this explanation in the caption of figure2, and/or in

the text where the figure is referenced, or at least say something like "relation between the change detection and painted areas is further explained in the discussion section". When I looked at figure 2 I tried to figure out what was going on with the patterns of erosion on the surface, and was perplexed until getting to this paragraph.

*We will add to the results section (where Figure 2F is cited first): "Relationships between these bedrock surface changes and paint erosion are further detailed in the discussion section.".*

Pg5 lines 24-27: So do the authors think these issuesăˇA ˇTair bubbles in the paint, and painting on wet rock surfacesăˇA ˇTaffected their measurements? It's a little unclear how much these factors may have influenced their results. Is it conjecture, or based on hindsight from their results?

*We actually did not see any air bubbles in our paint, but we mention this as an issue that could hypothetically lead to bias in the paint erosion patterns. Hence, to be clearer on this issue and to nevertheless point to the potential problem of air bubble inclusion, we will move the discussion of this problem to the section "General assessment of the erosion painting technique". There we will state: "The paint should be applied carefully (e.g., avoiding wet and dusty rock, and leaving sufficient time for drying), since incorporated air bubbles or insufficient drying could lead to shear detachment of the paint by flowing water alone, without abrasion of the surface.".*

Pg7 lines 9-12: I'm not sure how well this would necessarily work, because how well paint adheres to a surface is generally pretty sensitive to the surface characteristics. I would think that different rock types or compositions would typically have different "paintabilities". Its kind of hard to get paint to stick well to quartz, for example. In any case this effect should be acknowledged.

*Thank you for this hint! We will mention this when speaking about potential further studies with abrasion mills to provide the basis for more semi-quantitative studies of streamflow erosivities: "Laboratory tests (e.g., using the erosion mills of Sklar and Dietrich, 2001) could be used to explore the erodibility of different paints, the influence of applied paint thickness, and paint adhesion on different bedrock lithologies. Also the erosivity of the flow in the abrasion mill (i.e. its ability to erode the paint) could be studied by changing the mill's flow velocity or sediment loading. This analysis could serve as background for more semi-quantitative studies of natural flow erosivities.".*

Figure 2: In 2f and 2g, does "mm/2a" mean millimeters per two years? So is this not the amount of erosion measured between 4/6/2012 and 8/10/2013 (1 year and 4 months or so), but that amount of erosion, normalized up to two years? Please clarify what the 2 years means in this case.

*Yes, these erosion rates are derived from bedrock change detection measurements (2 million point measurements at mm resolution) over the time span 04 June 2012 – 08*

*October 2013. Since nearly all flushing events in 2012 and 2013 occurred in between these two laser scanning dates, the measured values represent erosion over the two years. We will mention this fact in the figure caption and in the results section. As already noted in the reply to the Anonymous reviewer above, we will report the erosion values as means over the two years of measurement, i.e. in mm/a.*

I realize the Lidar-based change detection measurements are not the main focus of the present work, but on the figure or in the main text I think some mention of uncertainty of these measurements is needed. Both uncertainty of the individual scans themselves, and also uncertainty in the differences.

*We will give an explanation on this topic in the section "Process inferences from erosion painting at the Gornera", as follows: "The uncertainty in the individual TLS change detection values was 2.2 mm over the biennial comparison, and thus was in the same order of magnitude as the detected change rates. However, the huge numbers of TLS measurements permit a stable general impression of surface changes, assuming their measurement errors are not spatially correlated (Beer et al., in review).".*

In 2e, it seems confusing to have the box and whisker plot rectangles not be centered over the time intervals that I think they're supposed to represent.

*We realized that it is more useful to give some statistical numbers on the flushing periods than showing additional boxplots. Hence, we will add a table with these numbers to the flushing height series in Figure 2 D and remove the boxplots (Figure 2 E).*

In figure 5c, I presume the dark vertical lines on either side of the white painted area is the rock being wet? Clarify in caption what the dark areas are. Conceivably could be intrusions or something.

*Thank you for pointing to this problem! Yes, the black stripes are wet rock sections; we will point this out in the figure caption.*

**C. Reply to the comments of Theodore Fuller**

This is a welcome addition to the literature on erosion in bedrock channels. One of the primary contributions of the paper is the presentation of field-based evidence that supports several hypotheses of bedrock channel erosion. The field observations presented here support at least three existing hypotheses on bedrock erosion processes: 1) bedrock erosion rates are highly variable depending on the in-channel orientation of bedrock surfaces; 2) lateral bedrock erosion via tools is focused at the base of the channel wall; 3) suspended sediment is capable of bedrock erosion.

> *Thank you for your kind recognition and résumé of our study.*

**Comments**

Field-based evidence of a substantial cross-channel gradient in sediment concentration during high, erosive channel flows is a valuable contribution to the literature. As such, I would like to see this explored in a little more detail (to the extent possible) perhaps by providing basic hydraulic conditions within and upstream of the gorge that might be driving this apparent concentration gradient.

> *Exploring the hydraulic conditions and the streambed's influence that drive this sediment routing certainly is worth a whole study (incorporating analysis of the various flushing events), but would go far beyond the scope of this paper. We do not have hard numbers on the hydraulics here, but in the discussion we will explain a bit more why we think the sediment is routed to the left side of the gorge: "Directly upstream of the inspected wall section (to the left of Figure 1 C), there are rock blocks of 2 m size in the streambed that leave a passage on the gorge's left side. This passage may channelize the sediment flow even when these blocks are submerged by the flushing water. Further, secondary currents due to turbulence induced by the boulders are also likely to have influenced the sediment distribution (Venditti et al., 2014). We do not have direct measurements of the spatial sediment transport distribution during the flushings, but the erosion painting technique was able to document the crucial influence of sediment routing in setting local erosion rates.".*

As a proof-of-concept study, I would have liked to see a more detailed discussion of the methodology. The authors briefly mention some of the potential pitfalls of the method in the discussion section but do not really address how these pitfalls can be avoided. Maybe the authors have indeed provided the necessary level of detail for a proof-ofconcept study, perhaps the editor has a better feeling for the detailed required for such a study.

> *As answered to the Anonymous reviewer above, we will add some more discussion of problems with the painting technique in the subsection "General assessment of the erosion painting technique", stating: "The paint should be applied carefully (e.g., avoiding wet and dusty rock, and leaving sufficient time for drying), since incorporated air bubbles or*

*insufficient drying could lead to shear detachment of the paint by flowing water alone,*
*without abrasion of the surface.".*

In light of the comment above, it might be prudent to focus a little less on the proofof-
concept idea, particularly in the abstract and introduction sections, and more on the
qualitative field-based evidence which lends support to several existing hypotheses.
Again, I think perhaps the most exciting thing here is that you have FIELD-BASED
evidence of patterns of bedrock erosion.

> *We agree, and we will avoid using the term "proof-of-concept" that often in the text, since*
> *we already discuss the findings of spatial bedrock erosion distribution and applicability of*
> *the technique in detail.*

More detail of the field site would help the reader put the findings in context. Estimate of
hydraulic conditions, grain size and mobility, primary lithology of channel boundaries: : :

> *We will add some info on the field site: "The local bedrock is serpentinite, and the bed*
> *sediment consists of both serpentinite and gneiss. […] Due to the characteristics of the*
> *flushing operations (i.e. short, steep hydrographs and evacuation of previously accumulated*
> *sediment), the mean transported bedload grain size ($D_{50}$) likely varies considerably during*
> *each flushing event and between flushing events. The $D_{50}$ of the natural streambed*
> *upstream of the hydropower water intake is ~4 cm at low flows, but it is unknown whether*
> *the average sediment load (including high flows) is finer or coarser than this. The sediment*
> *bed surface in the gorge returns to roughly the same height following each flushing event,*
> *but it likely varies strongly during the flushings themselves (Beer et al., in review).".*

**Line-by-line comments submitted as a supplemental document using the Adobe comment tool
within the manuscript.**

> *Thank you for your thorough commenting! We will gladly address your comments, as*
> *detailed in the following.*

Page 2 line 26: I realize the field site is described in detail in another paper but I think you should
at least give a brief description of bedrock lithology and a D50 of the bedload material.

> *We will add additional info on the field site as detailed in the answer just given above.*

Page 2 line 29: I think it should be made clear that the paint started at the sediment bed surface
and not the bedrock bed surface. Do you know the thickness of the alluvial deposit?

> *We will add a note to the methods, indicating that stripe painting started at the surface of*
> *the sediment bed: "… we painted several vertical stripes of 0.15 m width and 2.0 m height*
> *on two opposing straight and smooth bedrock walls, starting at the sediment bed surface*

*(Figure 1C and D; we unfortunately could not paint below the sediment surface due the water table in the sediment body).".*
*We assume that the alluvial deposit is several meters thick, since we have observed local m-deep scouring of the sediment (through the ponding water).*

Page 2 line 30: Do you have any sense of bed coverage during the actual flow events? Are all grain sizes mobile at flushing flows?

*We will add a note indicating that sediment bed height was relatively constant between flushing periods, but likely varied strongly during flushings: "The sediment bed surface in the gorge returns to roughly the same height following each flushing event, but it likely varies strongly during the flushings themselves (Beer et al., in review).".*
*Few times we observed parts of the sediment bed in the gorge completely scoured (after longer flushing events). So we assume all sediment sizes (except the big boulders) are mobile during flushings .*

Page 2 line 31: Why is a reference for a 'water table' altimeter important? Maybe this is just my own misunderstanding of the term used here. Is water table equivalent to the open channel water surface?

*We will change the term "water table altimeter" to "water surface altimeter" to make clear that this device monitored the height of the water surface in the gorge. Knowledge of this changing height is important for calculation of the submergence of the paint (Figures 2, 3 and 5).*

Page 3 line 7: Sounds like fantastic field site for bedrock erosion studies with distinct sediment transport events.

*This is the reason for our studies there ☺.*

Page 3 line 8: Could specify that 'flushing height'  is the same as flow depth (at least that is how I read this).

*Yes, we mean "flow depth" by "flushing height". We will change the text here and elsewhere to avoid this term, and will name the y-axis of Fig.2D "flushing event flow height", since we refer to the height of water stage in the gorge during the flushings.*

Page 3 line 17: At the risk of appearing to be a shameless plug for my own research, you could cite Fuller et al., 2016 (JGR-ES) which shows a similar result of decreasing erosion rates with height above the streambed.

*☺We will gladly cite Fuller et al., 2016, which was not yet published at the time we wrote this draft (so we originally referred to Fuller, 2014, PhD thesis).*

Page 4 line 30: Fuller et al., 2016, which just came out in May, is a more easily accessible reference than the dissertation cited here. The hypothesis suggested by Fuller et al. is one of bed load particles being deflected into the channel walls rather than falling through the water column.
*Ditto*

Page 5 line 26: I am glad to see this discussed as I was wondering about this for much of the paper. I would make a note of how painting on wet rock and drying time could affect results in the methods section of the paper just so the reader doesn't have questions in his/her mind as they read through the results and discussion.
*As detailed in the answers to both other reviewers above, we will better discuss problems with putting the paint on the rock in the section "General assessment of the erosion painting technique", stating: "The paint should be applied carefully (e.g., avoiding wet and dusty rock, and leaving sufficient time for drying), since incorporated air bubbles or insufficient drying could lead to shear detachment of the paint by flowing water alone, without abrasion of the surface.".*
*In the section on "Potential future applications of erosion painting" we will also note: "Laboratory tests (e.g., using the erosion mills of Sklar and Dietrich, 2001) could be used to explore the erodibility of different paints, the influence of applied paint thickness, and paint adhesion on different bedrock lithologies.".*

Page 5 line 29: At the authors discretion in their encouragement of this method...I would advocate for excavating all the way to the bedrock bed and starting the vertical stripes at that elevation. I think it's possible that erosion could be occurring on the wall below the surface of the alluvial bed...though from a practical standpoint, unless a field site is highly regulated like the current site the base of the bedrock wall will likely be below base level flows.
*We had the same idea of digging deeper in the sediment bed and painting the bedrock walls at lower elevations (below the sediment bed surface), since the sediment bed varies during the flushings. However, the water table in the sediment body is nearly flush with its surface and it was unfortunately not possible to paint below … We will mention this in the methods section: "To visualize variations of erosion with height above the streambed, we painted several vertical stripes of 0.15 m width and 2.0 m height on two opposing straight and smooth bedrock walls, starting at the sediment bed surface (Figure 1 C and D; we unfortunately could not paint below the sediment surface due the water table in the sediment body).".*

Page 6 line 29: Venditti et al., 2014 showed strong secondary flows, some oriented laterally toward the channel walls, as discharge enters a narrow bedrock canyon. You mention that the gorge measured here is only 30 m in length. Is it possible the type of flow described by Venditti et al., 2014 is contributing to the lateral transport of sediment particles? The Venditti paper should probably be referenced.

*Yes, secondary currents will likely have conditioned sediment transport in the gorge; we will refer to Venditti et al., 2014. We will extend our discussion: "The driving mechanism of this laterally focused sediment transport was probably the coarse boulder bed of the channel upstream of the gorge (Figure 1 B) that likely deflected the sediment flow. Directly upstream of the inspected wall section (to the left of Figure 1 C), there are rock blocks of 2m size in the streambed that leave a passage on the gorge's left side. This passage may channelize the sediment flow even when these blocks are submerged by the flushing water. Further, secondary currents due to turbulence induced by the boulders are also likely to have influenced the sediment distribution (Venditti et al., 2014). We do not have direct measurements of the spatial sediment transport distribution during the flushings, but the erosion painting technique was able to document the crucial influence of sediment routing in setting local erosion rates.".*

Page 7 line 3: when giving an example of the spatial distribution of sediment transport I would focus specifically on the cross-channel difference in sediment concentration that your results seem to support rather than generally mention the tools and cover effects.

*We will extend our notes on what the erosion painting technique was able to show in this study (at the beginning of "Potential future applications of erosion painting") and state: "Our results demonstrate that erosion painting is a straightforward method for (i) visualizing the spatial distribution of bedrock erosion (i.e. variations with position and orientation), for (ii) inferring the spatial distribution of sediment transport (i.e. the sediment tools and cover effects), and for (iii) localising the transient elevation of the sedimentary streambed under some circumstances.".*

Figure 2 comment 1: Interesting that even at heights above 2m where there were infrequent flows you still see erosion. If this is a real erosion signal and not just noise, it suggests erosion by some of the smaller particles in the distribution that may be in suspension. This would be field confirmation of the hypothesis put forth by Lamb et al., 2008 and Scheingross et al., 2014.

*Thank you for your encouragement with interpretation of the erosion mode with height above the bed! The detailed discussion of the spatial change detection values (e.g., erosion up to three meters above the streambed) is already provided in Beer et al., in review, and we should not repeat it here. For consistency with Figure 2 A-C, we will only show the erosion profile up to 2.1m above the bed in Figure 2 G.*

Figure 2 comment 2: Can you separate erosion according to discharge? If so, you could investigate the possible existence of an erosion threshold, like maybe no erosion occurs at the lower discharge events, maybe it is all due to the highest events. It could be that you don't have enough variation in flows due to controlled release events.

*No, we sadly cannot separate erosion according to discharge (or flushing height). We always had several different flushing events (differing in discharge and likely sediment transport) in between the re-paintings. At the beginning and at the end of the summer,*

*when there was much lower glacially driven discharge (and hence sediment transport), only a few small flushing events occurred in between the re-paintings and there was no paint erosion visible. In the summer with many high-flow events, we mostly had comparable erosion patterns (as detailed in Figure 3). So, we cannot determine a threshold of streamflow erosivity with our data. However, your hint is great, and we will gladly add this to the section on "Examples of advanced applications for field sites like the studied gorge would be (i) to more frequently check eroded paint patterns (e.g., after every erosive event) to find thresholds of paint erosion for constraining streamflow erosivity, (ii) to …".*